# Distribution of Superficial Body Temperature in Horses Ridden by Two Riders with Varied Body Weights

**DOI:** 10.3390/ani10020340

**Published:** 2020-02-21

**Authors:** Izabela Wilk, Elżbieta Wnuk-Pawlak, Iwona Janczarek, Beata Kaczmarek, Marta Dybczyńska, Monika Przetacznik

**Affiliations:** 1Department of Horse Breeding and Use, Faculty of Animal Sciences and Bioeconomy, University of Life Sciences in Lublin, Akademicka 13, 20–950 Lublin, Poland; izabela.wilk@up.lublin.pl (I.W.); iwona.janczarek@up.lublin.pl (I.J.); lisabravenhard@gmail.com (M.D.); moniczka81@onet.eu (M.P.); 2Department and Clinic of Animal Internal Diseases, Faculty of Veterinary Medicine, University of Life Sciences in Lublin, Akademicka 13, 20–950 Lublin, Poland; beatakaczmar1@gmail.com

**Keywords:** horse, thermography, heart rate parameters, rider weight

## Abstract

**Simple Summary:**

Leisure horse riding is becoming an increasingly popular activity, but often the regard for the welfare of recreational horses is insufficient. Most often, this situation is associated with workload or insufficient knowledge of the owners. Another problem may be overloading the animals with excessive weight of the rider, because the problem of obesity affects more and more societies. Carrying heavy loads significantly reduces horse comfort and affects movement mechanics. Prolonged use of the horse in this way may translate into health and lameness issues in the back and limbs. There are currently no strict weight limits for riders. However, we believe that horse users should pay more attention to this problem. As shown in this study, a horse’s load above 20% of his body weight, even with little effort, affects changes in surface temperature and the activity of the autonomic nervous system.

**Abstract:**

It was assumed that a horse with its rider body weight found in the upper limit may negatively impact the horse’s welfare. The objective of this paper was to analyze the differences in body temperature and selected heart rate parameters in horses in response to physical exercise accompanied by various rider’s body weight loads. The study was carried out on 12 leisure, 10–15-year-old warmblood geldings. The horses were ridden by two equally qualified riders whose body weights were about 20% and 10% of the average body weight (BW) of the animals (about 470 kg). Each rider rode each of the 12 horses for 13 min walking and 20 min of trotting. Images of the horse at rest, after physical exercise directly after unsaddling, and during the recovery phase (10 min after unsaddling) were taken with an infrared thermography camera. For analysis, the temperatures of selected body parts were measured on the surface of the head, neck, front, middle, and back (croup) parts of the trunk, forelimb, and hind limb. Immediately after the infrared thermography images were taken, the rectal temperature of the horse was measured. The heart rate parameters were measured at rest for 10 min directly before, during, and 10 min following the end of a training session. A multivariate analysis of variance (ANOVA) for repeated measurements was performed. Statistical significance was accepted for *p* < 0.05. A rider BW load on a horse of approximately 20% of the horse’s BW led to a substantial increase in the superficial temperatures of the neck, front, middle, and back parts of the trunk in relation to these body parts’ average temperatures when the load was about 10% BW. The head and limb average temperatures were not significantly affected by the load of the exercised horse. A horse’s load above 20% of his body weight, even with little effort, affects changes in surface temperature and the activity of the autonomic nervous system.

## 1. Introduction

Nowadays, obesity is a serious problem, also among riders [1]. The body weight (BW) of a rider is believed to have a major influence on the impact of physical exercise in horses [2]. As reported by Powell et al. [3], horses demonstrate no physiological changes while carrying 10–15% of their BW, which means about 50–75 kg of rider’s BW. Only when the load exceeds 25% of their BW do changes become visible, such as an increase in the basic physiological parameters and post-exercise muscle pain, as well as increased activity of the sympathetic nervous system [3,4]. Therefore, the results of the cited studies indicate that the load for a horse should not exceed 20%.

The BW of a rider set at 10% of a horse BW, believed to be optimal, is generally too low among the current horse rider population [5]. However, the BW of a rider should not be ignored, as is commonly done. Therefore, a recorded load of 10–15% has prompted a search for simple methods for a tailored selection of a horse for a given rider [6]. Moreover, modelling the seat based on the relation between the body weight of a rider to the body mass of a horse may be an interesting approach [7]. The selection of specially designed saddles may also be significant [8].

The aggravating impact of an excessive load of a rider’s BW increases together with increments in physical exercise to which the animal is subjected [9]. Greater rider weights require greater physical exertion from the horses, increasing the risk of an overload. All physical activity is associated with accelerated metabolism and the production of a significant amount of energy. Naturally, the body responds by increasing its temperature, which is proportional to the workload and results in 75% of the chemical energy generated during muscle work being transformed into heat energy and the rest being utilized as external work [10]. The general impact of increasing body temperature during physical exercise produces a reduction in the plasma volume and an increase in skin sweating and dilation of the blood vessels [11]. The increase in blood flow through the skin’s blood vessels enables so-called passive heat loss [12]. Passive, i.e., energy-saving, heat release by the body consists of heat emission in the form of infrared rays, taking heat from the body through the movement of air around the body and the heating of air particles near the skin [13].

Thermography is a tool that uses outside temperature measurement. It is used for lesion diagnosis in pre-diagnostic screening, such as identifying potential cases of inflammation musculoskeletal system, muscles and in the joints of large mammals. It gives a preliminary image of changes in surface body temperature, but detected irregularities require additional diagnostics. The same principle that is used for back pain diagnosis [14] and forelimb and back temperature on horses [15] can be used to determine if excessive BW of a rider causes an increase in the superficial body temperature of a horse. This method allows for quick, real-time diagnosis of temperature changes in individual parts of the body in response to exercise and lesions of tissues [16,17,18,19].

Discomfort sustained during riding also triggers a nervous response and changes the balance between sympathetic and vagal nervous system activity [20]. The analysis of heart rate and variability (HRV) is commonly applied to evaluate the sympathetic-vagal balance [21]. It has also been demonstrated that the excessive predominance of sympathetic system activity negatively influences the welfare and performance of horses [22]. Increased rMSSD (Root Mean Square of the Successive Differences) shows a shift towards a more parasympathetic dominance [21,23].

It is assumed that the weight of a rider’s body in the upper reference range might negatively impact the welfare of a horse, although some researchers did not show any relationship between the BW of a rider and a horse’s well-being [17,24]. The objective of this paper was to determine the changes in temperature distribution on the surface of the horse’s body in response to various rider weights using thermography as an imaging technique to assess muscle workload. The selected heart rate parameters (heart rate—HR and rMSSD) in horses were also analyzed to evaluate their nervous system response to various rider body weight loads.

## 2. Materials and Methods

### 2.1. Horses and Riders

The study was carried out on 12 saddle-ridden, 10–15-year-old warm-blood geldings. All of the horses had been kept for at least 12 months in the same stable in 3.5 × 3.5 m boxes with openwork partitions which enabled mutual eye contact. There was a manger in the corner of each box, an automatic drinking bowl, and a hay basket. Fodder was given to the horses three times daily, at 6.00 a.m., at noon, and at 6.00 p.m. The horses were given 2 kg of meadow hay and 1.5 kg of oats (three times a day) with mineral and vitamin supplements (once a day). The floor in the boxes was lined with wheat–rye straw once daily and was cleared every morning. The horses were regularly used for leisure riding for 1–2 h, six days per week, and spent the rest of the time in boxes or in paddocks and pastures depending on the season and weather conditions. The horses were weighed and measured at withers height the day preceding the start of the study (for details, see Table 1). The horses did not show symptoms of any disease or lameness.

Two professional riders of the same sex, unfamiliar to the studied horses, who usually take part in dressage intermediate level competitions and had equal qualifications, yet with different height and body mass, were invited to take part in the study. The percentage ratio of the rider’s body weight to the average body mass of the investigated horses was determined. The horse’s body weight was determined using animal weights BAKA-WAG (Bydgoszcz, Poland)-platform size 1.5 m × 2 m. The results are presented in Table 1.

### 2.2. The Experimental Design

Animal care and experimental procedures were in accordance with the European Committee Regulations on Protection of Experimental Animals and were approved by the Local Ethics Review Committee for Animal Experiments (no 27/2016).

The study was organized in the riding center when the horses were kept. The experiment was carried out in summer for two days with a one-day break in between the two data collection days, during which the horses were kept in a paddock. During the trial, the weather conditions were comparable (Table 2) and were taken from the official website: www.meteo.pl (archived on 12 November 2018).

The riders were assigned to ride the investigated horses in the following manner: on the first day, rider no. 1 rode the first group of six horses and rider no. 2 rode the second group of six animals. On day 2, rider no. 1 rode the second group of horses and rider no. 2 was assigned to the first group. The selection of the horses to the groups and the order of riding were random. Their coats were cleaned one hour before the trial. Before riding, the manes and tails of the horses were braid-plaited. This is the standard procedure for taking thermographic images for horses [25].

The horses were walked in pairs (one from each group) in halters into indoor arena, where measurements at rest were taken. Then were led onto a sanded, 25 × 50 m outdoor arena situated next to the stable. The riding ground was an open, uncovered area. The horses were saddled just before riding on the arena. Each horse had its own profiled cotton saddle pad, gel insulation, and leather saddle fastened with a neoprene girth. The saddles were initially tested for fitting with infrared thermography [26], and it was concluded, based on the measurements, that the saddles were fitted correctly for each of the animals. The riders used also typical dressage equipment, i.e., bridles with snaffle bit and reins, without any horsewhip, spurs, or other accessory equipment. The training session for experimental purposes consisted of 10 min of walking on a loose rein, 10 min of left-rising trotting, 3 min of walking, and 10 min of right-rising trotting. The average velocity of movement measured with a Polar GPS was 157 m/min (SD: 5.12) while walking and 218 m/min (SD: 8.35) trotting. The GPS sensors were placed with a wristband on the rider’s left arm. Following the ride, the horses were led by riders to the indoor arena, unsaddled, and they remained there until the end of the recovery phase.

## 3. Research Methods

### 3.1. Body Temperature Measurements in Horses

Images of the horses for body temperature measurements were acquired with a Fluke type Ti 9 infrared thermography camera (Fluke Corporation, Everett, Washington, USA). The measurements were taken in an indoor arena, which was well known to horses, situated between the outdoor arena and the stable. The indoor arena complied with the requirements for infrared thermography tests, i.e., it was closed and shadowed to prevent heat inflow from the outside such as solar radiation or airflow. In addition, it was draught-free with a stabilized ambient temperature of 21 °C. The relative air humidity was 45–48%. The thermal images were taken only in artificial light, which was situated at a distance of 6 m over the horse, under the ceiling of the indoor arena. The floor in the facility was even and hardened. The camera was located on a tripod, 150 cm from the ground. The horses were scanned from a distance of 3 m. Thermal images of the left side of the body were taken three times:(1)Directly before saddling (resting superficial body temperature),(2)Directly after unsaddling (post-exercise superficial body temperature),(3)10 min after unsaddling (recovery superficial body temperature).

The animals were accustomed to having thermal images taken of them.

The records were downloaded from a Secure Digital Card to a computer and then analyzed with a Fluke SmartView 4.3 (Eindhoven, The Netherlands). For analysis, the temperatures of selected body parts were measured on the surface of the head, neck, front, middle, and back (croup) parts of the trunk, forelimb, and hind limb. The final result was the average temperature of the whole body part, as outlined in Figure 1.

Each time, immediately after the infrared thermography images were taken, the rectal temperature of the horses was measured with an electronic KRUUSE Digi-Vet SC 12 thermometer.

### 3.2. Heart Rate Parameters

The heart rate parameters were measured at rest for 10 min directly before, during all training sessions, and for 10 min following the end of a training session (unsaddling was the starting point). The measurements were taken with Polar RS800CX devices (Polar Elektro Oy, Kempele, Finland). The monitor of the device was placed on the horses’ body with a rubber band around the girth and was on it throughout the measurements. The receiver was attached to the rubber band when the horse was unsaddled or to the saddle pad during training session. The data were then analyzed with PolarProTrainer 5.0 software (Kempele, Finland). The analyzed parameter was HR (heart rate frequency), the number of heart beats per minute. rMSSD (the root mean square of the successive differences between adjacent R-R intervals) within a short-period variability was also calculated. The rMSSD is the primary time-domain measurement used to estimate the high-frequency beat-to-beat variations that represent vagal regulatory activity. The animals were previously used to wearing heart rate monitors.

### 3.3. Statistical Methods

A multivariate analysis of variance (ANOVA) for repeated measurements was performed [27]. The rider factor (rider no. 1 (20% BW of horse); rider no. 2 (10% BW of horse)), subsequent measurement factor (at rest, post-exercise, recovery phase), and body part factor (head, neck, front of trunk, middle part of trunk, back of trunk, foreleg, hind leg) were considered. The significance of differences between the means was determined with a Tukey’s test. Significance of differences was calculated for all fixed factors and their interactions between factors. Statistical significance was accepted for *p* < 0.05.

## 4. Results

The considered factors and interaction between them were significant in the case of all parameters at the significance level, *p* < 0.05, except there was no effect of rider of horse rectal temperature (Table 3).

There was a significantly higher resting superficial body average temperature on the head, neck, and front part of the trunk compared to the average temperature of the front limb and hind limb in the horses before they were ridden by both riders (Table 4). Moreover, the average temperature of the middle part and back part of the trunk did not differ from the other body parts. Significant differences were not found in temperature for the analogous body parts in the horses before they were ridden by two different riders.

For the immediately post-exercise measurements, the average temperatures on the head and limb surface were significantly lower than the average temperatures of the other body parts (Table 5). The average superficial temperatures of the neck and each part of the trunk were significantly higher in the horses ridden by rider no. 1 (20% BW of horse) than in those ridden by rider no. 2 (10% BW of horse). Moreover, in horses ridden with a load of 10% BW, significant increases in the average temperatures of the front, middle, and back part of the trunk were found. The load of 20% BW induced significant increases in the average temperatures of the neck and all parts of the trunk compared to load of 10% BW (Figure 2).

Recovery average superficial temperature of the investigated body parts differed significantly between the individual parts (Table 6). For horses ridden by rider 20% BW of horse, the average temperature was significantly higher on the front part of the trunk compared to the other body segments. The lowest values were recorded for the head and limbs. In horses ridden by rider 10% BW of horse, the average temperatures on the head, middle part of trunk, croup, and limbs were significantly lower than on the neck and front of the trunk. Substantial differences between the average temperatures of the analogous parts of the body in the horses ridden by different riders referred to all three segments of the trunk and in each case, higher values were found in the horses ridden by rider 20% BW of horse (Table 5 and Table 6).

Post-exercise and recovery rectal temperatures in the horses ridden by both riders were significantly higher than the resting temperature (Table 7). Differences in rectal temperatures linked to the rider were not observed. HR values were significantly higher after work with rider no. 1 (20% BW of horse) than rider no. 2 (10% BW of horse) in both post-exercise and recovery phases. The rMSSD values, inversely, were significantly lower in horses ridden by rider no. 1 (20% BW of horse (Table 7).

## 5. Discussion

The results of the study indicate that the average superficial temperature on some, but not all, body parts of exercised horses varied despite the BW of the rider.

Differentiation of superficial body temperature is observed just at rest. The anterior body parts, i.e., the head, neck, and front part of the trunk, are warmer than the back part of the trunk and lower sections of the front and hind limbs [28]. Since a varied intensity of tissue blood perfusion is mainly responsible for a specific distribution of superficial body temperature [29], it thus would be even more interesting to know whether an analogous diversity also occurs in the horse during physical exercise and whether the body weight of carried riders affects the results.

Infrared thermography measurements taken directly after physical exercise showed a distribution of temperatures different than resting. Namely, the average temperature of the head surface was the lowest and was comparable with the average temperature of the peripheral limbs. In contrast, the average temperature of the neck surface approximates the head and limb temperature, yet only when the horses are loaded with 10% of their body weight. When the load increases more than twofold, the average temperature of the neck surface was elevated and comparable to the average temperature of the trunk. For the trunk, it should be emphasized that this segment of the body had the highest post-exercise average temperature, proportional to the load of the horse by the rider BW. Perhaps this situation may be caused by the harder work of the shoulder when being ridden at a higher load.

As to the impact of a rider’s body mass on the recorded results, it should be mentioned that the intensity of the physical exercise that the horses were subjected to might be, at most, compared to a routine warm-up before a core training session [30]. Even though the physical exercise was not intensive, the difference between the resting and post-exercise average temperatures of the neck and trunk with a load exceeding 20% was over 4 °C. As reported by Jodkowska et al. [28], a comparable increase in surface body temperature was reported in jumping horses directly after a jumping competition. These studies referred to more intense physical exercise, which implies that the current results are the consequence of loading the horse by BW of the rider, not the intensity of exercise.

The results of measurements of the recovery body temperature are also worth noting. Similar to the situation directly after physical exercise, a heavier rider BW is linked with significantly higher temperature values of the whole trunk. These findings mean that the outer body temperature does not decrease rapidly. It seems that if the regression of the body temperature is more pronounced, then a transitory increase is not so noticeable to the body. A slow recovery phase of body temperature emphasizes, even more, the need for thorough investigations on the impact of loads put on the horse on changes in the superficial body temperature because, as reported by Morgan et al. [31], a prolonged time of temperature restitution significantly increases oxygen consumption by the horse. Similar results were found for people. Borodulin-Nadzieja et al. [32] concluded that increased surface temperature in working miners increased the heart load by a slow normalization of HR, among others.

Though there is an evident impact of a rider’s BW on the increase of surface body temperature, it is significant that rectal temperature does not change substantially with heavier loads. The greater the rider weight, the greater the muscular effort, which results in more heat generated, which the body must carry away. Thermoregulation processes are designed to maintain the internal temperature at a constant level and remove excess heat to the outside. As reported by Marlin [33], the increased exercise which raised the internal body temperature to 42 °C should not present a risk to the health or life of the animals. The temperature recorded in the conducted study was lower than the threshold by ca 3.5 °C and, as such, should not be regarded as dangerous to the body.

It is commonly known that HR helps in assessing the response of the body to physical and mental workload [20,21]. A body which handles workload better has a lower activity of the sympathetic nervous system and lower HR. Furthermore, the HR recorded post-exercise in the horses ridden by the rider with heavier BW was still increased and was higher than in horses which were carrying 10% of their BW. According to Podolak et al. [34], HR approximating 120 bpm (which is comparable to the values recorded in the current study) after minor physical exercise is also a good indicator of the level of sympathetic nervous system activity. On the other hand, the rMSSD depicts the activity of a parasympathetic component of the autonomic nervous system, which has an inhibiting, i.e., relaxing, effect on the body. It has been found that this activity, both post-exercise and recovery, was much higher when the horse was loaded with a lighter weight, and higher than at rest. Thus, it may be concluded that minor physical exercise of horses loaded with a rider’s body mass equaling approximately 10% of its own BW may have a less aggravating impact on the horse’s body then loaded with a rider’s body mass about 20% of its own BW. It is worth noting that physiological responses, both body temperatures and HR were higher, and rMSSD was lower with the heavier rider’s BW, however, these responses were far from the maximal values noted in other exercised horses [21,28,34].

It would be worth extending the experiment to include several riders in the same weight category. This would allow checking whether, apart from the difference in body weight, there were other factors, e.g., individual riding style, that could affect the results. In addition, some authors did not note the effect of rider weight increase on physiological and behavioral parameters of horse [9]. This fact may indicate the complexity of factors that affect the horse’s response in response to increased workload. It should also be mentioned here the limitations associated with the use of thermography. This method requires comparable microclimate conditions in which pictures will be taken. Changing weather conditions or the season of the year can also affect the results.

In summary, the findings indicate that studies on the optimal rider’s body weight load on a horse should still be analyzed in a multifaceted manner. For now, it is worth emphasizing that a load of 20% significantly influences the horse’s body, which is demonstrated by the increased temperature of the neck and back, accompanied with the predominance of sympathetic nervous system activity, as compared to less loaded horses.

## 6. Conclusions

A rider BW load weighing approximately 20% of the horse BW led to a substantial increase in the superficial temperatures of the neck, front, middle, and back parts of the trunk in relation to these body parts’ temperatures when the load was about 10% BW. Moreover, the temperature of the head and limbs (especially) were not significantly affected by the load of the exercised horse. It should be noted that a load of 10% of the horse BW significantly influenced only the temperature of the back and positively affected the horse’s autonomic nervous system in relation to the resting measurements.

## Figures and Tables

**Figure 1 animals-10-00340-f001:**
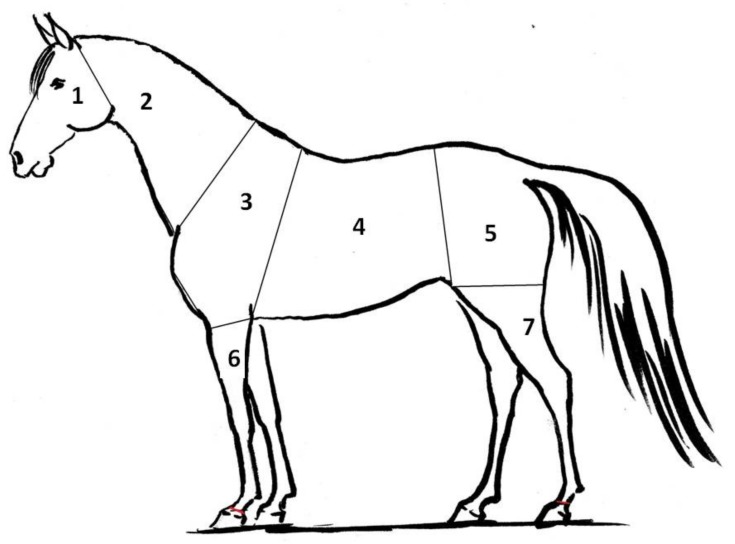
The seven areas of the body that were thermal imaged on 12 different horses: 1—head, 2—neck, 3—front of trunk, 4—middle part of trunk, 5—back of trunk, 6—foreleg, 7—hind leg.

**Figure 2 animals-10-00340-f002:**
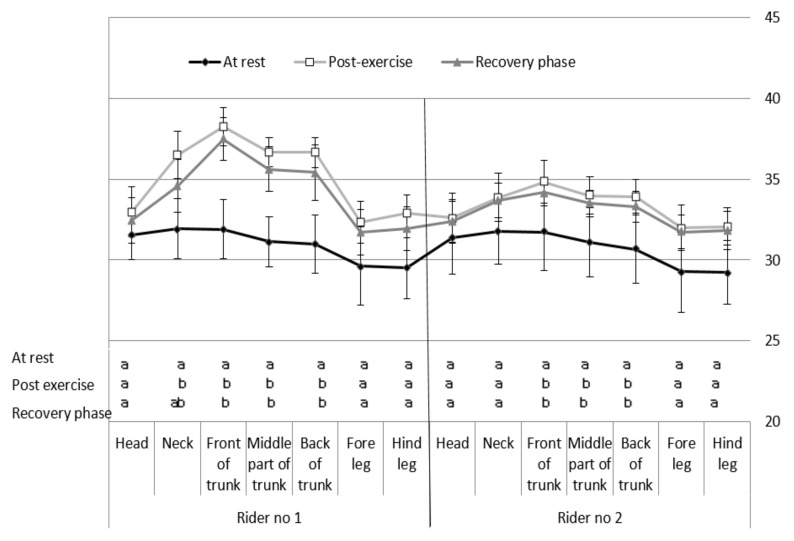
Average temperature (°C) of the surface on the investigated body parts of the horse recorded during subsequent stages of the trial. The means denoted with different letters differ significantly at *p* ≤ 0.05.

**Table 1 animals-10-00340-t001:** Biometric profiles of the horses and riders involved in the study.

Horses		Rider No 1		Rider No 2	
Average height at the withers ± SD (cm)	Average body weight ± SD (kg)	Body weight with saddle (kg)/% body weight of horse	Height (cm)	Body weight with saddle (kg)/% body weight of horse	Height (cm)
163.4 ± 2.11	469.4 ± 32.8	100/21.3	178	50/10.6	158

**Table 2 animals-10-00340-t002:** Weather conditions during the experiment.

Day of the Study	Description and Air Temperature (°C)	Relative Air Humidity (%)	Wind Speed (m/s)	Atmospheric Pressure (hPa)
First day	Cloudy, 21	45	2	1022
Second day	Cloudy, 22	47	2	1018

**Table 3 animals-10-00340-t003:** Sources of variation.

Sources of Variation	Df	F	*p*
Superficial body temperatures			
rider	1	17.02	0.0194
body part	6	11.78	0.0000
subsequent measurement	2	250.73	0.0000
rider * body part	6	13.96	0.0000
subsequent measurement * rider	2	167.21	0.0315
subsequent measurement * body part	12	2.67	0.0021
rider * body part * subsequent measurement	12	4.94	0.0078
Rectal body temperature			
rider	1	0.46	0.5057
subsequent measurement	2	21.21	0.0084
rider * subsequent measurement	2	19.7	0.0394
HR			
rider	1	6.03	0.0378
subsequent measurement	2	40.76	0.0000
rider * subsequent measurement	2	23.28	0.0000
rMSSD			
rider	1	4.72	0.0457
subsequent measurement	2	8.45	0.0009
rider * subsequent measurement	2	5.17	0.0399

df—degrees of freedom, F—value of F-snedecor test, *p*—probability value; * mark interactions between factors; factors significant at *p* < 0.05, highly significant at *p* < 0.01.

**Table 4 animals-10-00340-t004:** Resting average superficial temperatures (°C) of the studied body parts of the horse (Means ± SD).

Body Part	Head	Neck	Front of Trunk	Middle Part of Trunk	Back of Trunk	Foreleg	Hind Leg
Rider no 1 20% BW	31.6 ± 1.55 a	31.9 ± 1.86 a	31.9 ± 1.83 a	31.1 ± 1.54 ab	31.0 ± 1.79 ab	29.6 ± 2.44 b	29.5 ± 1.91 b
Rider no 2 10% BW	31.4 ± 2.06 a	31.8 ± 2.03 a	31.7 ± 2.37 a	31.1 ± 2.15 ab	30.7 ± 2.13 ab	29.3 ± 2.03 b	29.2 ± 1.98 b

The means denoted with different letters (a, b—in rows,) differ significantly at *p* ≤ 0.05.

**Table 5 animals-10-00340-t005:** Post-exercise average superficial temperatures (°C) of the studied body parts of the horse (Means ± SD).

Body Part	Head	Neck	Front of Trunk	Middle Part of Trunk	Back of Trunk	Foreleg	Hind Leg
Rider no 1 20% BW	33.0 ± 1.43 ax	36.5 ± 1.62 bx	38.3 ± 1.30 cx	36.7 ± 1.40 bx	36.7 ± 1.71 bx	32.4 ± 1.41 ax	32.9 ± 1.37 ax
Rider no 2 10% BW	32.6 ± 1.35 ax	32.9 ± 1.05 ay	34.8 ± 0.85 by	34.0 ± 0.81 by	33.9 ± 0.96 by	32.0 ± 1.05 ax	32.1 ± 1.19 ax

The means denoted with different letters (a, b, c—in rows, x, y—in columns) differ significantly at *p* ≤ 0.05.

**Table 6 animals-10-00340-t006:** Recovery phase average superficial temperatures (°C) of the studied body parts of the horse (Means ± SD).

Body Part	Head	Neck	Front of Trunk	Middle Part of Trunk	Back of Trunk	Foreleg	Hind Leg
Rider no 1 20% BW	32.5 ± 1.53 ax	34.6 ± 1.46 bx	37.5 ± 1.18 cx	35.6 ± 0.88 bx	35.4 ± 0.92 bx	31.7 ± 1.29 ax	31.9 ± 1.13 ax
Rider no 2 10% BW	32.4 ± 1.52 ax	33.7 ± 1.49 bx	34.2 ± 1.33 by	33.5 ± 1.14 aby	32.4 ± 1.06 ay	31.7 ± 1.41 ax	31.8 ± 1.16 ax

The means denoted with different letters (a, b, c—in rows, x, y—in columns) differ significantly at *p* ≤ 0.05.

**Table 7 animals-10-00340-t007:** Rectal temperature (°C) and heart rate parameters in the studied horses (Means ± SD).

Body Temperature (°C)					HR (bpm)				rMSSD (ms)		
	At rest	Post-exercise	Recovery phase	*p*	At rest	Post-exercise	Recovery phase	*p*	At rest	Post-exercise	Recovery phase
Rider no 1 20% BW	37.2 ± 0.58 ax	38.9 ± 0.61 bx	38.7 ± 0.66 bx	0.054	39.3 ± 4.97 ax	117.4 ± 11.6 bx	67.4 ± 8.63 cx	0.009	105.0 ± 104.2 ax	114.0 ± 83.9 ax	107.2 ± 94.6 ax
Rider no 2 10% BW	37.4 ± 0.71 ax	38.6 ± 0.57 bx	38.4 ± 0.64 bx	0.057	41.8 ± 10.4 ax	81.8 ± 11.6 by	48.7 ± 7.39 ay	0.062	104.0 ± 153.9 ax	224.0 ± 106.9 by	222.6 ± 101.9 by

The means denoted with different letters (a, b, c—in rows, x, y—in columns) differ significantly at *p* ≤ 0.05.

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
