# Peer review of "Distribution of Superficial Body Temperature in Horses Ridden by Two Riders with Varied Body Weights"

_animals, 2020, doi:10.3390/ani10020340_

Round 1

Reviewer 1 Report

There is growing concern about the weight that horses are asked to carry as obesity in riders becomes more commonplace. thus this research is timely and presents some useful information. However, the manuscript can be much improved by the inclusion and reference to other research done in this area - please consult other published research. Additionally, there are no statistics reported in the results. which detracts from the value of the results. Please see itemized comments below:

L13 – consider replacing “care” with “regard for the welfare of…” to read more clearly

L14 – delete “too much” before “…overloading the animals” to read more clearly

L15 – replace “overweight” with “obesity”

L16-18 – consider rewriting for clarity: “Carrying heavy loads significantly reduces horse comfort and affects movement mechanics. Prolonged use of the horse in this way may translate into health and lameness issues in the back and limbs.”

L19 – replace “horse users” with “horse riders” since you are referring specifically to riders

L25 – replace “rider’s” with “rider”

L28 – “The study was carried out over two days…”

L29-30 – please indicate when after physical exercise and when during the recovery phase the images were taken. Also clarify what parts of the horse’s body were imaged.

L30-31 – “during the thermography phase…” do you mean during the riding and recovery phase? Can you clarify what heart rate parameters were recorded?

I know one is always limited by word count in the abstract, but the abstract could be written more scientifically to convey what was actually measured, how, when, and what measured were analyzed. Is there a concluding sentence to the abstract that can summarize your findings? The simple summary should also reflect the same findings – ie. the simple summary implies there were changes in heart rate but the abstract does not mention this.

L39-40 – this is a very strange opening sentence that does not have any supporting reference. How does increased body mass relate to increased social well-being? Why does that have any bearing on horsemanship? Would that mean that all obese people are interested in horses?

L41 – consider replacing “intensity” with “impact”, as physical exercise can be intense with a lightweight jockey.

L41 – references should appear in numerical order, not alphabetically numbered. Thus, the Pagan reference should be [1]. Please correct all reference numbers.

L42 – reads “horses demonstrate a slightly increased physical exercise while carrying 10% to 15% of their BW” according to Powell’s study. However, Powell did not evaluate a 10% carrying load (only 15, 20, 25 and 30%). Also, what does “increased physical exercise” mean? Powell evaluated work rate (watts), heart rate, respiration rate, rectal temperature, plasma lactate and serum creatine kinase. Furthermore, they found no differences in any of these measures until the carrying load was 25% or greater. So for clarity, perhaps this sentence should read “horses demonstrate no physiological changes while carrying 10% to 15% of their BW”

L44-45 – you mention behavioural changes, but neither Powell nor Ille reported any behavioural measures, only physiological. Furthermore, you mention an increase in basic physiological parameters without reference to what these might be.

L53-54 – the reference to Autio does not seem appropriate here where you are discussing rider BW increases and physical exercise, since Autio used thermography to measure body heat loss.

L54 – “rider weights” not rider’s weights

L70 – since von Schweinitz and Soroko et al used thermography for back pain diagnosis and forelimb and back temperatures on various horses respectively, it would be better to say that based on previous research on thermography in horses, it is a likely tool to measure changes in body temperature of horses being ridden by riders of different body weights. This would be more clear than what you have written which implies that those two studies looked at rider body weight.

L73 – delete the word “saddle” and indicate that changes in nervous response and sympathetic and vagal activity is in the horse

L75 – delete “of heart rate” after (HRV)

L77 – delete the word “value” after performance

L77-78 – this line could bear more description. What exactly is rMSSD and why is it a useful measure?

L83 – which heart rate parameters did you select? Please be specific.

L101 – “rider’s” not riders’

L119 – is there a significance to braiding the horses manes and tails?

L123 – “cotton saddle pad” not horse pads

L124 – “leather saddle” not saddles

L127 – rather than stating “aversive tackle” which implies an opinion, perhaps use “accessory equipment”

L130 – you mention Polar GPS but have not indicated when or how you equipped the horses with the Polar monitor.

L131 – clarify “hand walked”

L138 – what do you mean by “restitution”?

L142 – please specify the distance the artificial light was from the horse

L143-144 – does this mean that the camera lens was located 150cm from the ground?

L144 – why did you only image the left side of the horse’s body and not both sides?

L146-148 – can you indicate when the horses were taken from the outdoor arena to the indoor arena for each of the three imaging sessions?

L149 – “The animals were accustomed to having thermal images taken of them.”

L152 – how exactly were the body temperature measurements taken? Was it an average of the whole body part as outlined in Fig 1?

L155 – caption for Figure 1 should indicate that these were the seven areas of the body that were thermal imaged on 12 different horses. Also, part 4 is mislabelled.

L157 – was the rectal temperature taken after each of the three thermography sessions?

L160 – exactly when during the training session were the heart rate parameters measured? Was it the same time frame for each horse?

L172 – do you mean a Tukey’s test? A t-test is different from Tukey’s.

L177 – here and everywhere where you have presented results – please include the test statistics (eg. F-statistic and degrees of freedom) and the p-values.

L182 – please expand the table caption to be more informative. A reader looking at this may interpret the body temp readings to be that of the riders. Are these readings in °C? Please state. Also, standard deviations should have the same number of decimal places as the mean (1 in this case). Further, is there a reason to present individual rows for the riders – you state that there were no differences in superficial resting body temperature between the riders, so it would be acceptable to average the means for each body part.   

L184 – clarify that the results in this paragraph for immediately post-exercise

L185 – do you mean the average superficial temperatures of the neck, etc?

L187 – it is confusing when you switch between rider number and load percentage when reporting the results based on rider. Can you be consistent either in referring to rider number or load percentage (load percentage would be more clear).

L191 – delete

L193-194 – please elaborate this figure caption. The reader should be able to fully understand the figure without having to read the entire paper.

Perhaps there is something missing from your figure – what do each of the points and the vertical line represent? Is this data from the two different riders?

L196 and 198 – indicate that these are the average temperatures of the body parts for each rider

L198-199 – you have written “horses ridden by rider no. 2, the temperatures on the head, croup and limbs were significantly lower than on the neck and front of the trunk”, but Table 5 shows that temperatures on head, middle, croup and limbs were lower.

L203 and 205 and 212 – please expand caption as mentioned previously. Please indicate that temperatures are in °C.

Table 6 – for consistency it would be good to have the Rider no in the left-hand column like the other tables.

L231 – “when being ridden at a higher load.”

L270 –the assessment that the lighter rider actually had a relaxing effect on the horse is questionable. I think that your statements in the paragraph below (L274-276) may be more reflective of what is occurring.

L274-276 – it is good that you noted that multiple riders could produce different results based on other factors than just weight. Similarly, you should discuss the limitations of using thermography.

L290 – delete the 1. before Reference and add an ‘s’ (References)

References – there should be a ; between author names, journal titles should be in italics. Please provide doi’s where available.

There are numerous other references on thermography and rider weight in horses that could be consulted, for example:

Christensen J.W., Uldahl M. Did you put on weight? The influence of increased rider weight on horse behavioural and physiological parameters. Proceedings of the 15th International Society for Equitation Science, Guelph, ON. 2019. Cebulj-Kadunc, Nina ; Frangez, Robert ; Zgajnar, Jaka ; Kruljc, Peter. Cardiac, respiratory and thermoregulation parameters following graded exercises in Lipizzaner horses/Utjecaj intenziteta vjezbe na srcani i disni sustav te tjelesnu temperaturu konja lipicanske pasmine. Veterinarski Arhiv, 2019, Vol.89(1), p.11(13) Åšpitalniak-Bajerska, Kinga ; Zaborski, Daniel ; Poźniak, BÅ‚ażej ; Dudek, Krzysztof ; Janczarek, Iwona. Exercise-induced changes in skin temperature and blood parameters in horses. Archiv fuer Tierzucht, 2019, Vol.62(1), pp.205-213 Priego Quesada, Jose Ignacio. Editor. Application of Infrared Thermography in Sports Science Soroko, Maria ; Howell, Kevin. Infrared Thermography: Current Applications in Equine Medicine. Journal of Equine Veterinary Science, January 2018, Vol.60, pp.90-96.e2 Gunnarsson, V ; Stefánsdóttir, G. J ; Jansson, A ; Roepstorff, L. The effect of rider weight and additional weight in Icelandic horses in tölt: part II. Stride parameters responses. Animal, 2017, Vol.11(9), pp.1567-1572 Matsuura, A ; Sakuma, S ; Irimajiri, M ; Hodate, K. Maximum permissible load weight of a Taishuh pony at a trot. Journal of animal science, August 2013, Vol.91(8), pp.3989-96

Some of these studies do not find any differences in physiological parameters even with quite high loads. Please be sure to comment on this.

Author Response

Reviewer 1

There is growing concern about the weight that horses are asked to carry as obesity in riders becomes more commonplace. thus this research is timely and presents some useful information. However, the manuscript can be much improved by the inclusion and reference to other research done in this area - please consult other published research. Additionally, there are no statistics reported in the results. which detracts from the value of the results. Please see itemized comments below:

Authors

Dear Reviewer, thank you very much for all relevant comments that helped improve our manuscript

Reviewer

L13 – consider replacing “care” with “regard for the welfare of…” to read more clearly

L14 – delete “too much” before “…overloading the animals” to read more clearly

L15 – replace “overweight” with “obesity”

L16-18 – consider rewriting for clarity: “Carrying heavy loads significantly reduces horse comfort and affects movement mechanics. Prolonged use of the horse in this way may translate into health and lameness issues in the back and limbs.”

L19 – replace “horse users” with “horse riders” since you are referring specifically to riders

L25 – replace “rider’s” with “rider”

L28 – “The study was carried out over two days…”

L29-30 – please indicate when after physical exercise and when during the recovery phase the images were taken. Also clarify what parts of the horse’s body were imaged.

L30-31 – “during the thermography phase…” do you mean during the riding and recovery phase? Can you clarify what heart rate parameters were recorded?

I know one is always limited by word count in the abstract, but the abstract could be written more scientifically to convey what was actually measured, how, when, and what measured were analyzed. Is there a concluding sentence to the abstract that can summarize your findings? The simple summary should also reflect the same findings – ie. the simple summary implies there were changes in heart rate but the abstract does not mention this.

Authors

All the tips above have been included in the text

Reviewer

L39-40 – this is a very strange opening sentence that does not have any supporting reference. How does increased body mass relate to increased social well-being? Why does that have any bearing on horsemanship? Would that mean that all obese people are interested in horses?

Authors

the sentence has been corrected

Reviewer

L41 – consider replacing “intensity” with “impact”, as physical exercise can be intense with a lightweight jockey.

Authors

the sentence has been corrected

Reviewer

L41 – references should appear in numerical order, not alphabetically numbered. Thus, the Pagan reference should be [1]. Please correct all reference numbers.

Authors

the list of references has been improved

Reviewer

L42 – reads “horses demonstrate a slightly increased physical exercise while carrying 10% to 15% of their BW” according to Powell’s study. However, Powell did not evaluate a 10% carrying load (only 15, 20, 25 and 30%). Also, what does “increased physical exercise” mean? Powell evaluated work rate (watts), heart rate, respiration rate, rectal temperature, plasma lactate and serum creatine kinase. Furthermore, they found no differences in any of these measures until the carrying load was 25% or greater. So for clarity, perhaps this sentence should read “horses demonstrate no physiological changes while carrying 10% to 15% of their BW”

L44-45 – you mention behavioural changes, but neither Powell nor Ille reported any behavioural measures, only physiological. Furthermore, you mention an increase in basic physiological parameters without reference to what these might be.

Authors

the sentence has been corrected

Reviewer

L53-54 – the reference to Autio does not seem appropriate here where you are discussing rider BW increases and physical exercise, since Autio used thermography to measure body heat loss.

Authors

the reference has been corrected

Reviewer

L54 – “rider weights” not rider’s weights

Authors

the sentence has been corrected

Reviewer

L70 – since von Schweinitz and Soroko et al used thermography for back pain diagnosis and forelimb and back temperatures on various horses respectively, it would be better to say that based on previous research on thermography in horses, it is a likely tool to measure changes in body temperature of horses being ridden by riders of different body weights. This would be more clear than what you have written which implies that those two studies looked at rider body weight.

Authors

the sentence has been corrected

Reviewer

L73 – delete the word “saddle” and indicate that changes in nervous response and sympathetic and vagal activity is in the horse

L75 – delete “of heart rate” after (HRV)

L77 – delete the word “value” after performance

Authors

the sentence has been corrected

Reviewer

L77-78 – this line could bear more description. What exactly is rMSSD and why is it a useful measure?

L83 – which heart rate parameters did you select? Please be specific.

L101 – “rider’s” not riders’

Authors

the description has been corrected

Reviewer

L119 – is there a significance to braiding the horses manes and tails?

Authors

this is the standard procedure for taking thermographic images for horses (Soroko and Morel, 2016)

Reviewer

L123 – “cotton saddle pad” not horse pads

L124 – “leather saddle” not saddles

L127 – rather than stating “aversive tackle” which implies an opinion, perhaps use “accessory equipment”

Authors

the sentence has been corrected

Reviewer

L130 – you mention Polar GPS but have not indicated when or how you equipped the horses with the Polar monitor.

Authors

the description has been corrected

Reviewer

L131 – clarify “hand walked”

L138 – what do you mean by “restitution”?

L142 – please specify the distance the artificial light was from the horse

L143-144 – does this mean that the camera lens was located 150cm from the ground?

Authors

the description has been corrected

Reviewer

L144 – why did you only image the left side of the horse’s body and not both sides?

Authors

we chose only the left side of the body based on the research of other authors who did not notice differences between the left and right side of the horse's body (e.g Jodkowska et al. 2011)

Reviewer

L146-148 – can you indicate when the horses were taken from the outdoor arena to the indoor arena for each of the three imaging sessions?

Authors

the description has been corrected in part ‘’The experimental design’’

Reviewer

L149 – “The animals were accustomed to having thermal images taken of them.”

Authors

the sentence has been corrected

Reviewer

L152 – how exactly were the body temperature measurements taken? Was it an average of the whole body part as outlined in Fig 1?

L155 – caption for Figure 1 should indicate that these were the seven areas of the body that were thermal imaged on 12 different horses. Also, part 4 is mislabelled.

Authors

the description of figure 1 has been corrected

Reviewer

L157 – was the rectal temperature taken after each of the three thermography sessions?

L160 – exactly when during the training session were the heart rate parameters measured? Was it the same time frame for each horse?

Authors

the description has been corrected

Reviewer

L172 – do you mean a Tukey’s test? A t-test is different from Tukey’s.

L177 – here and everywhere where you have presented results – please include the test statistics (eg. F-statistic and degrees of freedom) and the p-values.

Authors

the description of statistical methods has been corrected. An additional table is included in the results chapter, which describes the p-values, F and df.

Reviewer

L182 – please expand the table caption to be more informative. A reader looking at this may interpret the body temp readings to be that of the riders. Are these readings in °C? Please state. Also, standard deviations should have the same number of decimal places as the mean (1 in this case). Further, is there a reason to present individual rows for the riders – you state that there were no differences in superficial resting body temperature between the riders, so it would be acceptable to average the means for each body part.   

Authors

we left the values for both riders to indicate precisely that at rest there were no differences between the body temperature of the horses.

Reviewer

L184 – clarify that the results in this paragraph for immediately post-exercise

L185 – do you mean the average superficial temperatures of the neck, etc?

L187 – it is confusing when you switch between rider number and load percentage when reporting the results based on rider. Can you be consistent either in referring to rider number or load percentage (load percentage would be more clear).

L191 – delete

Authors

the description has been corrected

Reviewer

L193-194 – please elaborate this figure caption. The reader should be able to fully understand the figure without having to read the entire paper.

Perhaps there is something missing from your figure – what do each of the points and the vertical line represent? Is this data from the two different riders?

Authors

the figure has been corrected

Reviewer

L196 and 198 – indicate that these are the average temperatures of the body parts for each rider

Authors

the description has been corrected

Reviewer

L198-199 – you have written “horses ridden by rider no. 2, the temperatures on the head, croup and limbs were significantly lower than on the neck and front of the trunk”, but Table 5 shows that temperatures on head, middle, croup and limbs were lower.

Authors

In the description we included only fundamentally different from each other, the middle part of trunk was similar to both warmer and colder parts of the body

Reviewer

L203 and 205 and 212 – please expand caption as mentioned previously. Please indicate that temperatures are in °C.

Table 6 – for consistency it would be good to have the Rider no in the left-hand column like the other tables.

L231 – “when being ridden at a higher load.”

Authors

the description has been corrected

Reviewer

L270 –the assessment that the lighter rider actually had a relaxing effect on the horse is questionable. I think that your statements in the paragraph below (L274-276) may be more reflective of what is occurring.

Authors

the sentence has been corrected

Reviewer

L274-276 – it is good that you noted that multiple riders could produce different results based on other factors than just weight. Similarly, you should discuss the limitations of using thermography.

Authors

description has been added

Reviewer

L290 – delete the 1. before Reference and add an ‘s’ (References)

Authors

has been corrected

Reviewer

References – there should be a ; between author names, journal titles should be in italics. Please provide doi’s where available.

There are numerous other references on thermography and rider weight in horses that could be consulted, for example:

Christensen J.W., Uldahl M. Did you put on weight? The influence of increased rider weight on horse behavioural and physiological parameters. Proceedings of the 15th International Society for Equitation Science, Guelph, ON. 2019. Cebulj-Kadunc, Nina ; Frangez, Robert ; Zgajnar, Jaka ; Kruljc, Peter. Cardiac, respiratory and thermoregulation parameters following graded exercises in Lipizzaner horses/Utjecaj intenziteta vjezbe na srcani i disni sustav te tjelesnu temperaturu konja lipicanske pasmine. Veterinarski Arhiv, 2019, Vol.89(1), p.11(13) Åšpitalniak-Bajerska, Kinga ; Zaborski, Daniel ; Poźniak, BÅ‚ażej ; Dudek, Krzysztof ; Janczarek, Iwona. Exercise-induced changes in skin temperature and blood parameters in horses. Archiv fuer Tierzucht, 2019, Vol.62(1), pp.205-213 Priego Quesada, Jose Ignacio. Editor. Application of Infrared Thermography in Sports Science Soroko, Maria ; Howell, Kevin. Infrared Thermography: Current Applications in Equine Medicine. Journal of Equine Veterinary Science, January 2018, Vol.60, pp.90-96.e2 Gunnarsson, V ; Stefánsdóttir, G. J ; Jansson, A ; Roepstorff, L. The effect of rider weight and additional weight in Icelandic horses in tölt: part II. Stride parameters responses. Animal, 2017, Vol.11(9), pp.1567-1572 Matsuura, A ; Sakuma, S ; Irimajiri, M ; Hodate, K. Maximum permissible load weight of a Taishuh pony at a trot. Journal of animal science, August 2013, Vol.91(8), pp.3989-96

Some of these studies do not find any differences in physiological parameters even with quite high loads. Please be sure to comment on this.

Authors

Thank you for pointing out useful references

Reviewer 2 Report

Interesting experiment and important topic in many equestrian disciplines. One of the main limitation of the study as it is mentioned in the discussion is the low number of horses and riders in order to increase the power of the results. 

Please consider the following points: 

Line 31: is that real that with thermography you can measure the internal temperature?

Line 40: add reference

Line 49-50: add reference

Line 63-64: add reference

Line: please describe rMSSD acronym (it is explained in line 164, but it should be here as this is the first time the work comes up in the article).  

Line 98: I do recommend to change the sentence to: “The horses did not show…”

Line 113: Since this is a temperature study, it would have been better to use the wet-bulb temperature measurements instead of the meteo website

Line 116: delete “the other”

Line 130: I would use km/hr for the speed, most of the equine population are most familiar with km/hr than m/min

Line 162: was the Polar device attached to the horse all the time? Even when the thermography images were taken?

Line 252: The internal temperature is not really a central venous temperature taken with a probe, is rather a rectal temperature taken in a single time and as demonstrated in previous work by Dr. David Marlin, the rectal temperature taken immediately after exercise is not a reflection of the internal temperature (central venous temperature).

Line 259-273: I would add that training a horse with the same rider, the horse will get used to the body weight of the rider and perhaps not influence as much those differences seen in the study

Author Response

Reviewer 2

Interesting experiment and important topic in many equestrian disciplines. One of the main limitation of the study as it is mentioned in the discussion is the low number of horses and riders in order to increase the power of the results. 

Please consider the following points: 

Reviewer

Line 31: is that real that with thermography you can measure the internal temperature?

 Authors

the sentence has been corrected

Reviewer

Line 40: add reference

Line 49-50: add reference

Line 63-64: add reference

 Authors:

references have been added

Reviewer

Line: please describe rMSSD acronym (it is explained in line 164, but it should be here as this is the first time the work comes up in the article).  

 Authors

description has been added

Reviewer

Line 98: I do recommend to change the sentence to: “The horses did not show…”

 Authors

the sentence has been corrected

Reviewer

Line 113: Since this is a temperature study, it would have been better to use the wet-bulb temperature measurements instead of the meteo website

 Authors

This is a very accurate tool that was recommended in consultation with meteorologists

Reviewer

Line 116: delete “the other”

Authors

the sentence has been corrected

 Reviewer

Line 130: I would use km/hr for the speed, most of the equine population are most familiar with km/hr than m/min

  Authors

m/min are commonly used in equestrian sports

Reviewer

Line 162: was the Polar device attached to the horse all the time? Even when the thermography images were taken?

Authors

description has been added

 Reviewer

Line 252: The internal temperature is not really a central venous temperature taken with a probe, is rather a rectal temperature taken in a single time and as demonstrated in previous work by Dr. David Marlin, the rectal temperature taken immediately after exercise is not a reflection of the internal temperature (central venous temperature).

 Authors

the description has been changed to rectal temperature

 Reviewer

Line 259-273: I would add that training a horse with the same rider, the horse will get used to the body weight of the rider and perhaps not influence as much those differences seen in the study

Authors

this was not the purpose of our paper

Round 2

Reviewer 1 Report

Thank you to the authors for substantial improvement of this manuscript. There are still many small formatting details that should be easily addressed. In particular, I would suggest displaying results either in table format or in figure format, but not both since they both show the same information. Also please pay close attention to proper formatting for references. It would also improve your paper to refer in your discussion to the other papers I indicated in my last review as references. Some of these papers do not find an effect of rider weight on the horse, and it would be important to acknowledge that and provide some plausible explanations. This would strengthen your own paper.

Please see the following specific comments.

L14 – delete “too much” as overloading already implies this

L25 – delete “over two days” to reduce word count. It is not important to include here

L28 – reword to "Each rider rode each of the 12 horses for 13 minutes walking..."

L38 and 39 – include p-values where relevant

L40-41 - this concluding sentence is not implied from the information above. Perhaps better to state like you have in the simple summary "a horse's load above 20% of his body weight, even with little effort, affects changes in surface temperature and the activity of the autonomic nervous system."

L45 – delete “overweight or” and simply refer to obesity as the problem. You still need a reference for this statement.

L75-76 - this is a sentence fragment. Either incorporate it into the previous sentence, or rewrite as a complete sentence. EG. "The same principle that is used for back pain diagnosis [13] and forelimb and back temperature on horses [14] can be used to determine if excessive BW of a rider causes an increase in the superficial body temperature of a horse."

L88 – correct to horse’s body

L104 – correct to the horses did not show…

L107 – correct to the rider’s body weight…

L117 – state that the one-day break was inbetween the two data collection days

L119 – no need to include reference 24 in your reference list. You have included the url in the text which is sufficient. Also include your justification for this as you indicated in your response to reviewers. "this is the standard procedure for taking thermographic images for horses (Soroko and Morel, 2016)."

L126 – you indicate the horses were brought directly to the outdoor arena but below you mention that thermography took place immediately before saddling, and that was in the indoor arena.

L136 – how was the GPS placed on the rider’s hand? Were the GPS sensors glued to the rider's hand, or worn on a wristband or held by the rider??

L137 – who carried the horses? Reword to “the horses were directed… or the horses were led…”

L138 – “the time of the measurements” is very vague. Can you reword this perhaps to recovery phase?

L141 – remove the word “of” between “measurements” and “were acquired”

L144 - Suggest deleting this line since thermography measurements also occurred before being ridden.

L149 – indicate that the lights were attached to the ceiling of the indoor arena (rather than the roof)

L163 – the number 5 is missing from Fig 1 caption

L167 – which 10min during the exercise did you analyze the HR data from? This is important information

L169-171 - could you be more specific with how the HR monitor was attached to the horse? EG "An elasticized surcingle containing the transmitter electrodes was placed around the girth of the horse, under the saddle. The receiver was attached ???..."

When exactly was the HR monitor placed on the horse?

L179 – your abstract says rider no 1 was 21.3% of BW. Please correct throughout manuscript. Also close your bracket after “10% BW of horse))…”

L179 - subsequent measurement factor is an odd term and not very descriptive. Can you call is something like time point?

L180 - n is generally used to indicate the number of subjects, not the number of factors. Simply list the factors

L182 – clearly indicate that the significance of differences was calculated for all fixed factors and their interactions...

L186-187 - remove bolding. Reword to be grammatically correct "...p<0.05, except there was no effect of rider of horse rectal temperature."

L188 - Table 3 is helpful, but perhaps could be better organized to be more clear. I would suggest using another column on the left for each aspect you are comparing (superficial body temp, rectal body temp, HR, rSSMD) and then grouping the rows appropriately.

The caption for Table 3 also needs expanding to clearly explain what the table is showing.

L191 - in all cases throughout the manuscript, more correct to write average superficial body temp..., not superficial body average temp

L198 - since there is no y, delete this or remove the x's altogether. Your sentence in the text already indicates there is no difference.

L201 – what you have called Table 4 is now actually Table 5. Renumber your tables from this point on

L204 and 205 - increases compared to what? Resting temps?

L207 - Fig 2 is more clear now, but I think it was good when you had a solid vertical line between Rider 1 and 2. Please label the y-axis with units. The letters along the bottom are okay although you should clearly indicate that each row of letters corresponds with a specific time point. Also, your text indicates that there are differences between the riders, but your figure doesn't show this. It would be more helpful to have a complete picture.

Caption could still be more comprehensive (eg. include n for horses and riders).

This figure shows the same information as you have presented in your tables. Data should be presented only in one format - either a table or a figure.

L210 – your other tables have Rider no 1 and Rider no 2. Keep consistent whichever way you choose

L214 - you are still interchanging rider no with rider weight %. It makes it difficult for the reader to follow which rider is which. Please stay consistent throughout.

L217 – there is a decrease in average temp for middle as well as head, croup and limbs according to Table 5 (which should be Table 6)

L227 – delete the word factor after rider

L228 – close the bracket after (10% BW of horse)

L231 - use decimal instead of comma for all p-values in table

L243 – change to “…temperatures different than resting…”

L261 – correct to heavier rider BW

L268 – delete comma after Borodulin-Nadzieja et al.

L297 – correct to “It should also be mentioned…”

L313 - need to standardize formatting of references throughout. See instructions to authors. https://www.mdpi.com/authors/references. In particular, use abbreviated journal name throughout. Remove second period after last author's name in many instances. DOI should be in format DOI:10.xxx.xxx. Make sure all years are in bold. Add page numbers 1-15 for ref 5. Use correct referencing for conference proceedings (#8) and texts (#10, 29)

Author Response

Reviewer:

Thank you to the authors for substantial improvement of this manuscript. There are still many small formatting details that should be easily addressed. In particular, I would suggest displaying results either in table format or in figure format, but not both since they both show the same information. Also please pay close attention to proper formatting for references. It would also improve your paper to refer in your discussion to the other papers I indicated in my last review as references. Some of these papers do not find an effect of rider weight on the horse, and it would be important to acknowledge that and provide some plausible explanations. This would strengthen your own paper.

Authors:

Dear Reviewer, thank you again for all your valuable comments. They have undoubtedly helped to improve our manuscript.

Reviewer:

Please see the following specific comments.

L14 – delete “too much” as overloading already implies this

L25 – delete “over two days” to reduce word count. It is not important to include here

L28 – reword to "Each rider rode each of the 12 horses for 13 minutes walking..."

L38 and 39 – include p-values where relevant

L40-41 - this concluding sentence is not implied from the information above. Perhaps better to state like you have in the simple summary "a horse's load above 20% of his body weight, even with little effort, affects changes in surface temperature and the activity of the autonomic nervous system."

Authors:

all comments have been taken into account

Reviewer:

L45 – delete “overweight or” and simply refer to obesity as the problem. You still need a reference for this statement.

Authors:

Has been corrected, reference has been added

Reviewer:

L75-76 - this is a sentence fragment. Either incorporate it into the previous sentence, or rewrite as a complete sentence. EG. "The same principle that is used for back pain diagnosis [13] and forelimb and back temperature on horses [14] can be used to determine if excessive BW of a rider causes an increase in the superficial body temperature of a horse."

Authors:

the sentence has been corrected

Reviewer:

L88 – correct to horse’s body

L104 – correct to the horses did not show…

L107 – correct to the rider’s body weight…

L117 – state that the one-day break was inbetween the two data collection days

Authors:

Has been  corrected

Reviewer:

L119 – no need to include reference 24 in your reference list. You have included the url in the text which is sufficient. Also include your justification for this as you indicated in your response to reviewers. "this is the standard procedure for taking thermographic images for horses (Soroko and Morel, 2016)."

Authors:

references have been corrected

Reviewer:

L126 – you indicate the horses were brought directly to the outdoor arena but below you mention that thermography took place immediately before saddling, and that was in the indoor arena.

L136 – how was the GPS placed on the rider’s hand? Were the GPS sensors glued to the rider's hand, or worn on a wristband or held by the rider??

L137 – who carried the horses? Reword to “the horses were directed… or the horses were led…”

L138 – “the time of the measurements” is very vague. Can you reword this perhaps to recovery phase?

L141 – remove the word “of” between “measurements” and “were acquired”

L144 - Suggest deleting this line since thermography measurements also occurred before being ridden.

L149 – indicate that the lights were attached to the ceiling of the indoor arena (rather than the roof)

L163 – the number 5 is missing from Fig 1 caption

L167 – which 10min during the exercise did you analyze the HR data from? This is important information

Authors:

all information was included in the text, sentences were corrected

 Reviewer:

L169-171 - could you be more specific with how the HR monitor was attached to the horse? EG "An elasticized surcingle containing the transmitter electrodes was placed around the girth of the horse, under the saddle. The receiver was attached ???..."

When exactly was the HR monitor placed on the horse?

Authors:

the description has been completed

Reviewer:

L179 – your abstract says rider no 1 was 21.3% of BW. Please correct throughout manuscript. Also close your bracket after “10% BW of horse))…”

L179 - subsequent measurement factor is an odd term and not very descriptive. Can you call is something like time point?

L180 - n is generally used to indicate the number of subjects, not the number of factors. Simply list the factors

L182 – clearly indicate that the significance of differences was calculated for all fixed factors and their interactions...

L186-187 - remove bolding. Reword to be grammatically correct "...p<0.05, except there was no effect of rider of horse rectal temperature."

Authors:

the description has been completed

Reviewer:

L188 - Table 3 is helpful, but perhaps could be better organized to be more clear. I would suggest using another column on the left for each aspect you are comparing (superficial body temp, rectal body temp, HR, rSSMD) and then grouping the rows appropriately.

The caption for Table 3 also needs expanding to clearly explain what the table is showing.

Authors:

after consultation, our statisticians maintain that this is the standard and most readable way to describe the analysis of variance

Reviewer:

L191 - in all cases throughout the manuscript, more correct to write average superficial body temp..., not superficial body average temp

L198 - since there is no y, delete this or remove the x's altogether. Your sentence in the text already indicates there is no difference.

L201 – what you have called Table 4 is now actually Table 5. Renumber your tables from this point on

L204 and 205 - increases compared to what? Resting temps?

Authors:

the description has been corrected

Reviewer:

L207 - Fig 2 is more clear now, but I think it was good when you had a solid vertical line between Rider 1 and 2. Please label the y-axis with units. The letters along the bottom are okay although you should clearly indicate that each row of letters corresponds with a specific time point. Also, your text indicates that there are differences between the riders, but your figure doesn't show this. It would be more helpful to have a complete picture.

Caption could still be more comprehensive (eg. include n for horses and riders).

This figure shows the same information as you have presented in your tables. Data should be presented only in one format - either a table or a figure.

Authors: we made some adjustments to make the figure more readable. Fig 2 shows a comparison between subsequent stages of the trial. These comparisons are not included in the tables, that's why we'd like to keep this figure.

Reviewer:

L210 – your other tables have Rider no 1 and Rider no 2. Keep consistent whichever way you choose

L214 - you are still interchanging rider no with rider weight %. It makes it difficult for the reader to follow which rider is which. Please stay consistent throughout.

L217 – there is a decrease in average temp for middle as well as head, croup and limbs according to Table 5 (which should be Table 6)

L227 – delete the word factor after rider

L228 – close the bracket after (10% BW of horse)

L231 - use decimal instead of comma for all p-values in table

L243 – change to “…temperatures different than resting…”

L261 – correct to heavier rider BW

L268 – delete comma after Borodulin-Nadzieja et al.

L297 – correct to “It should also be mentioned…”

Authors:

the description has been corrected

Reviewer:

L313 - need to standardize formatting of references throughout. See instructions to authors. https://www.mdpi.com/authors/references. In particular, use abbreviated journal name throughout. Remove second period after last author's name in many instances. DOI should be in format DOI:10.xxx.xxx. Make sure all years are in bold. Add page numbers 1-15 for ref 5. Use correct referencing for conference proceedings (#8) and texts (#10, 29)

 Authors:

the references has been corrected

This manuscript is a resubmission of an earlier submission. The following is a list of the peer review reports and author responses from that submission.